# A Korean Cattle Weight Prediction Approach Using 3D Segmentation-Based Feature Extraction and Regression Machine Learning from Incomplete 3D Shapes Acquired from Real Farm Environments

Chang Gwon Dang [1], Seung Soo Lee [1], Mahboob Alam [1], Sang Min Lee [1], Mi Na Park [1], Ha-Seung Seong [1], Min Ki Baek [2], Van Thuan Pham [2], Jae Gu Lee [1,*] and Seungkyu Han [2,*]

1   National Institute of Animal Science, Rural Development Admission, Cheonan 31000, Republic of Korea
2   ZOOTOS Co., Ltd., R&D Center, Anyang 14118, Republic of Korea
*   Correspondence: jindog2929@korea.kr (J.G.L.); lion@zootos.com (S.H.); Tel.: +82-10-4030-1929 (J.G.L.); +82-10-6674-6886 (S.H.)

**Abstract:** Accurate weight measurement is critical for monitoring the growth and well-being of cattle. However, the traditional weighing process, which involves physically placing cattle on scales, is labor-intensive and stressful for the animals. Therefore, the development of automated cattle weight prediction techniques assumes critical significance. This study proposes a weight prediction approach for Korean cattle using 3D segmentation-based feature extraction and regression machine learning techniques from incomplete 3D shapes acquired from real farm environments. Firstly, we generated mesh data of 3D Korean cattle shapes using a multiple-camera system. Subsequently, deep learning-based 3D segmentation with the PointNet network model was employed to segment 3D mesh data into two dominant parts: torso and center body. From these segmented parts, the body length, chest girth, and chest width of Korean cattle were extracted. Finally, we implemented five regression machine learning models (CatBoost regression, LightGBM, polynomial regression, random forest regression, and XGBoost regression) for weight prediction. To validate our approach, we captured 270 Korean cattle in various poses, totaling 1190 poses of 270 cattle. The best result was achieved with mean absolute error (MAE) of 25.2 kg and mean absolute percent error (MAPE) of 5.85% using the random forest regression model.

**Keywords:** 3D segmentation; feature extraction; regression machine learning; weight estimation

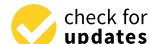

## 1. Introduction

The agricultural sector plays a pivotal role in meeting human food needs, with livestock farming serving as a vital source of meat, milk, and related products. To effectively manage and promote sustainable livestock production, the accurate weight estimation of livestock holds a critical position. Traditional livestock weighing methods often involve labor-intensive processes that cause stress to the animals, consequently negatively impacting overall productivity.

The development of an algorithm for livestock weight estimation without direct contact with the animals is essential. This approach addresses ethical concerns by the minimizing stress and discomfort of the animals during the weighing process.

In today's era of rapid technological advancements, the integration of computer vision-based techniques in livestock farming aligns perfectly with the movement towards smart and precision agriculture. The application of computer vision in livestock weight estimation represents a significant leap forward for automation and data-driven decision-making in agriculture.

Leveraging this technology, farmers can gain real-time insights into livestock weight and health, allowing for more efficient resource allocation, early disease detection, and

improved breeding strategies. This has the potential to increase productivity, reduce costs, and contribute to sustainable agricultural practices.

The development of algorithms to predict livestock weight through computer vision-based techniques has garnered significant attention, with various research studies in this area. There are two main approaches: 2D (two-dimensional) image analysis and 3D (three-dimensional) image analysis.

First, we review 2D image analysis approaches. Tasdemir and Ozkan [1] conducted a study to predict the live weight of cows using an artificial neural network (ANN) approach. They captured cows from various angles, applied photogrammetry to calculate body dimensions, and predicted live weight using ANN-based regression. Anifah and Haryanto [2] proposed a fuzzy rule-based system to estimate cattle weight, extracting body length and circumference as features to feed the fuzzy logic system for weight estimation. Ana et al. [3] conducted a study to predict live sheep weight using extracted features and machine learning. They captured sheep images from top view, created masks of the top view, and measured six distances in the mask as features to feed a random forest regression model. Weber et al. [4] proposed a cattle weight estimation approach using active contour and regression trees bagging. They first segmented the images, then created a hull from the segmented images, then extracted features, and predicted weight using a random forest model.

Compared to 2D image analysis approaches, 3D image processing approaches have gained more research attention in recent years. Jang et al. [5] estimated body weight for Korean cattle using 3D images, capturing them from the top view. After extracting body length, body width, and chest width, they built a linear function to calculate cattle weight. Na et al. [6] proposed a solution to predict cattle weight using depth images, capturing images from the top view, segmenting them, and extracting the characteristics of shape and size for cattle weight prediction using machine learning model. Kwon et al. [7] reconstructed a pig 3D model, created distances along pig's body as features, and utilized neural networks to predict pig weight. Hou et al. [8] collected data using LIDAR (light detection and ranging) sensor, segmented 3D beef object models using PointNet++ [9], measured body length and chest girth, and calculated weight using a pre-defined formula. Ruchay et al. [10] proposed a model for predicting live weight based on augmenting 3D point clouds in the form of flat projections and images with regression deep learning. Na et al. [11] developed a pig weight prediction system using Raspberry Pi, capturing RGB-D (Red Geen Blue Depth) images from the top view of pigs, extracting body characteristics and shape descriptors after segmenting the images, and applying various regression machine learning models to predict pig weight. Le et al. [12] calculated body sizes, surface area, length, and morphological traits from completed 3D shapes acquired using a laser scanning device to feed into a regression model for dairy cow weight estimation. Cominotte et al. [13] captured top view 3D images of cattle, extracted features from segmented images, and used linear and non-linear regression models to predict beef cattle weight. Martins et al. [14] also captured top view and side view 3D images, measuring several distances to feed into the Lasso regression model for body weight estimation.

Both the 2D analysis and 3D analysis approaches exhibit distinct advantages and drawbacks. The advantage of the 2D analysis approaches is that 2D imaging offers ease in segmentation and measurement processing, leveraging existing technology. Additionally, 2D images can be used to measure perimeter and area morphology, which also are features in the model to predict cattle weight. However, its limitation lies in the absence of depth information when using a single camera, constraining certain morphological measurements [10]. For example, chest girth (chest circumference) is replaced by chest diameter and chest depth measurements. The limitations of the 2D method can be overcome with 3D analysis approaches by using 3D cameras, but the cost is too high and the 3D data processing processes are often more complicated, so they are still not widely utilized.

Whether employing 2D or 3D image analysis approaches, a common formula involves the extraction of features for subsequent weight prediction. However, the feature extraction

process often relies on 2D segmented images or projection masks of 3D images, which can make it challenging to accurately represent 3D spatial elements, such as chest girth. Research has shown that chest girth is a critical factor in weight calculation [15].

In this study, we propose an approach that extracts features based on 3D segmentation, enabling us to measure features with precision, incorporating 3D spatial elements accurately. Furthermore, while previous studies were primarily conducted in laboratory or fence environments, our research predicts weights using 3D shapes acquired from real farm environments.

The main contributions of this proposal are as follows:

1. We introduce an effective approach for predicting Korean cattle weight using vision-based techniques.
2. We present a straightforward method for extracting cattle body dimensions through 3D segmentation.
3. We explore multiple regression machine learning algorithms for Korean cattle weight prediction.
4. Our approach not only predicts Korean cattle weight but also automatically measures three essential body dimensions of the cattle, facilitating further analysis.

## 2. Materials and Methods

### 2.1. Data Acquisition

To collect 3D Korean cattle data, we designed a specialized multiple-camera system, which is illustrated in Figure 1. In Figure 1a, you can see the system's design, and in Figure 1b, you can observe the actual setup.

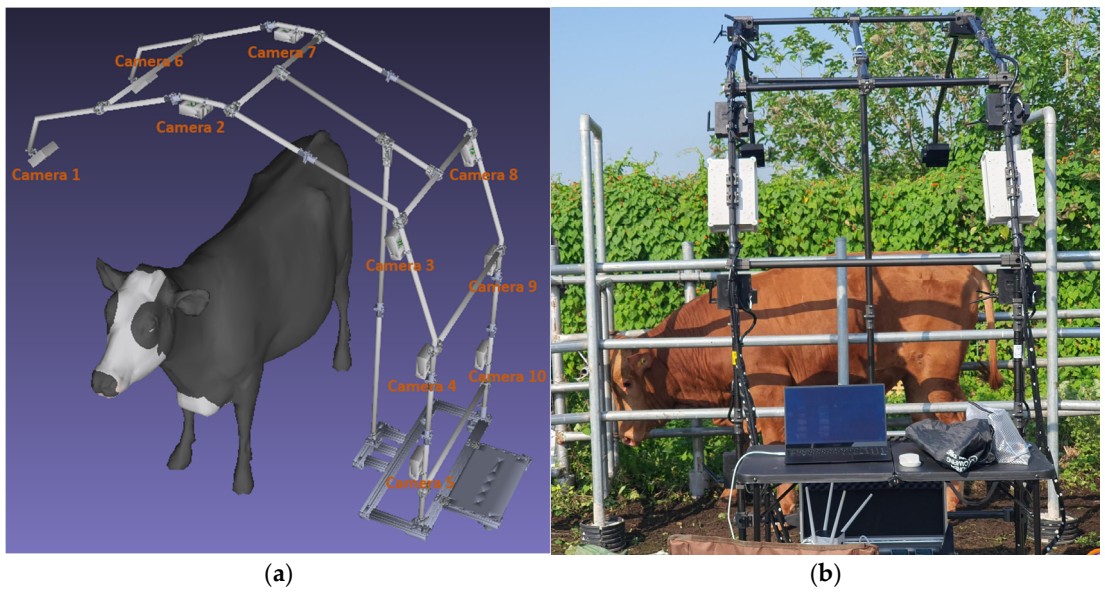

(**a**)　　　　　　　　　　　　　　　　　(**b**)

**Figure 1.** Multiple-camera system. (**a**) System design; (**b**) real-world deployment.

The system comprises ten stereo cameras arranged in a half-ring configuration, maximizing the coverage of Korean cattle as they pass by. Ideally, an image acquiring system should form a symmetrical U-shape to capture data from all angles. However, practical considerations, such as bulkiness, mobility issues, and animal fear, make such a design unfeasible. Our mechanical design, in contrast, is lightweight, flexible, and collapsible when not in use. This approach ensures efficient data acquisition without causing distress to the livestock. The mechanical components' sizes were specifically chosen to encompass the full range of Korean cattle, accommodating heights from 1 m to 1.5 m and lengths spanning from 1.5 m to 2.3 m. These design parameters ensure the adaptability of our

system to cater to the cattle between 8 to 24 months. Information on camera specification is shown in Table 1.

**Table 1.** Camera specifications.

| Device | Specifications |
|---|---|
| Depth Camera (Intel Realsense D435i) | Use environment: Indoor/Outdoor<br>Baseline [mm]: 50<br>Resolution: 1920 × 1080 px<br>Frame rate: 30 fps<br>Sensor FOV [1]<br>$(H \times V \times D):\ 69.4° \times 42.5 \times 77(\pm 3)$<br>User environment: Indoor/Outdoor<br>Connection: USB-C 3.1 |

[1] FOV: field of view.

The relative translation and rotation between all cameras remained constant throughout the data collection process. We employed stereo cameras, allowing each camera to capture two infrared images: a left infrared image and a right infrared image. Figure 2 provides an example featuring ten left infrared images from our proposed camera system. The ten right images captured by the system exhibit similar characteristics.

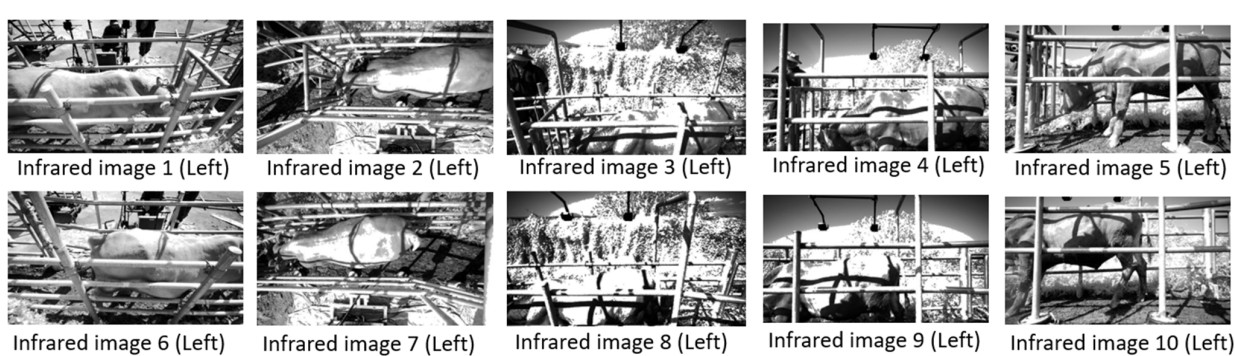

Infrared image 1 (Left) Infrared image 2 (Left) Infrared image 3 (Left) Infrared image 4 (Left) Infrared image 5 (Left)

Infrared image 6 (Left) Infrared image 7 (Left) Infrared image 8 (Left) Infrared image 9 (Left) Infrared image 10 (Left)

**Figure 2.** Left infrared images from our capturing system.

We generated 3D data from the left and right images of each camera using stereo matching, as described in [16]. Because the pre-defined relative distances and rotation angles of the cameras are constant, we could align the 3D images from each individual camera. The 3D mesh data was then reconstructed using the Poisson surface reconstruction algorithm [17] to construct a comprehensive 3D representation of the entire scene featuring the Korean cattle.

Once this was completed, we subtracted the background scene, resulting in the creation of the 3D mesh data only containing the Korean cattle, as exemplified in Figure 3. In Figure 3, each row displays the left view, top view, and right view of an animal. To assess the accuracy of our reconstruction process, we randomly and manually measured the length and chest girth of 10 Korean cattle and compared with the reconstructed results, revealing a length error of less than 1% and a chest girth error of 2.3%. Considering the intended application of reconstruction for weight prediction, these errors fall within acceptable ranges. Notably, the mesh data on the right side and the under area of the cattle appears incomplete due to our system's flexible design, which is designed to adapt to the real-world farm environments.

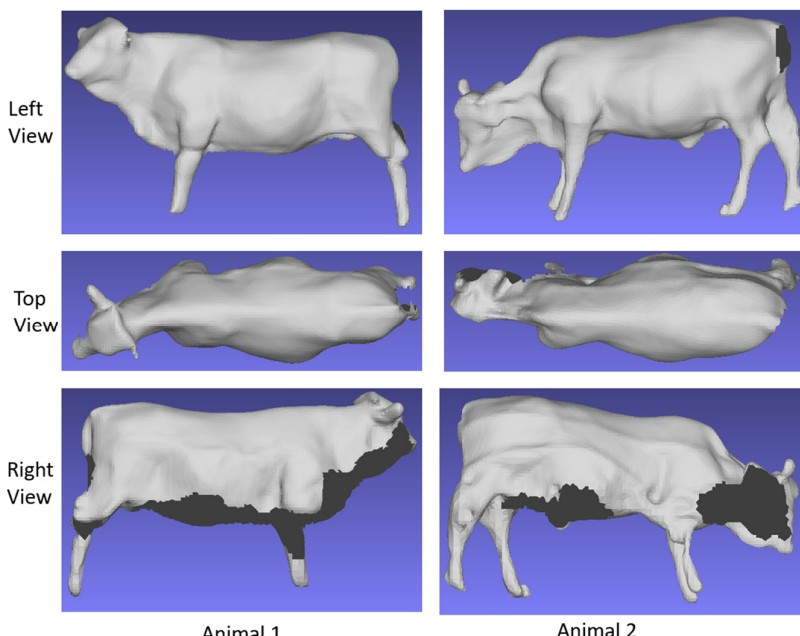

**Figure 3.** Three-dimensional mesh data of two random animals (Korean cattle) after reconstruction.

We conducted data collection on two separate occasions, in August 2023 and September 2023, at two distinct farms located in Seosan province, South Korea. Our dataset consists of a total of 270 cattle, ranging in age from 9 months to 12 months. For each individual animal, we captured between 3 to 5 shots in various poses, resulting in a collection of 1190 3D data files. Concurrently, we recorded the weight of each animal during the data capture process. The weight of the cattle in our dataset varies within the range of 300 kg to 600 kg. The weight distribution is visualized in Figure 4.

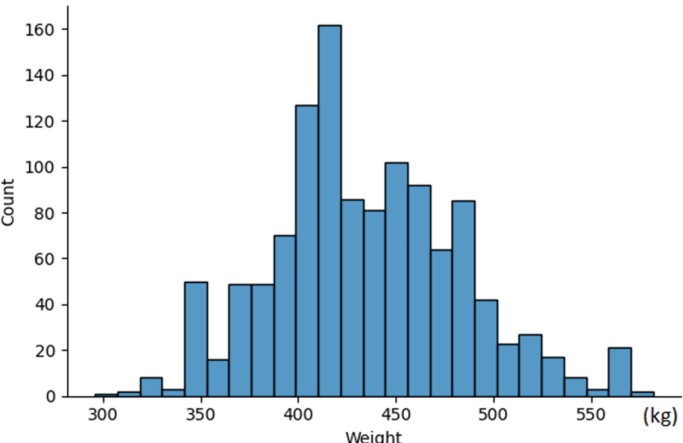

**Figure 4.** Weight distribution of Korean cattle used in this study.

### 2.2. Proposed Pipeline Overview

The overall diagram of the proposed pipeline is depicted in Figure 5. After the reconstruction process, the 3D images of Korean cattle were saved as 3D mesh files. The 3D mesh data were sampled into multiple point cloud data for the 3D segmentation process. Two segmentation models are designed for this project: torso segmentation and center body segmentation.

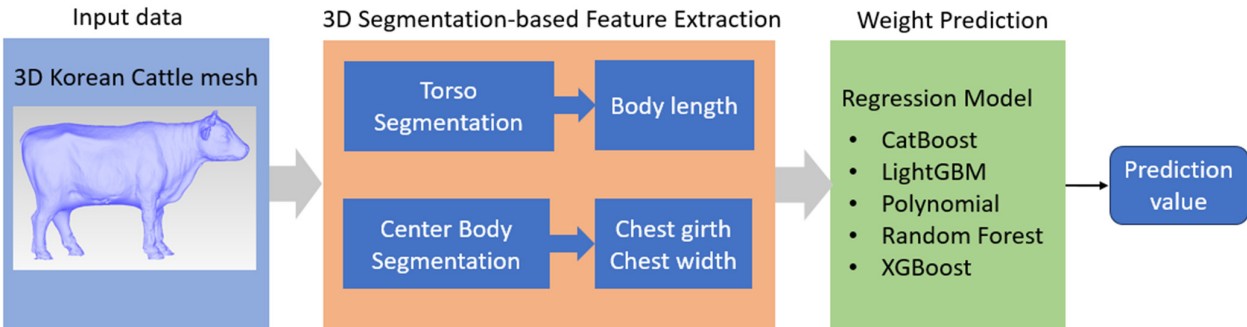

**Figure 5.** Overview structure diagram of proposed pipeline.

The output of the torso segmentation was used to measure the body length, while the output of the center body segmentation was used to extract the chest girth and chest width. After three important body dimensions are extracted, a regression machine learning model was developed to predict Korean cattle weight with these three dimensions as input. We applied five of the most prominent regression machine learning models: CatBoost, LightGBM (light gradient boosting machine), polynomial, random forest, and XGBoost (extreme gradient boost).

### 2.3. Three-Dimensional Segmentation-Based Feature Extraction

2.3.1. Definition of Korean Cattle Body Dimensions

A recent study [15] has demonstrated the feasibility of determining cattle weight by measuring 10 specific distance parameters which are chest girth, body length, chest width, rump width, hip height, wither height, pelvic width, rump length, chest depth, and hip bone width. These parameters were ranked based on their influence ratios on weight. In this study, we propose a solution that automates the measurement of three body dimensions, which have shown the highest influence ratios on weight. Subsequently, we utilized these dimensions to develop our weight prediction model. These three critical body dimensions are body length, chest girth, and chest width. Detailed measurement definitions for these body dimensions are provided in Table 2, and the corresponding measurement sites for each body dimension are visually represented in Figure 6.

**Table 2.** Definitions of body dimensions for Korean cattle.

| Body Dimensions | Symbol | Definition |
|---|---|---|
| Body length | BL | Horizontal length of the body |
| Chest girth | CG | Perimeter of the vertical body axis at the chest |
| Chest width | CW | Maximum width of chest |

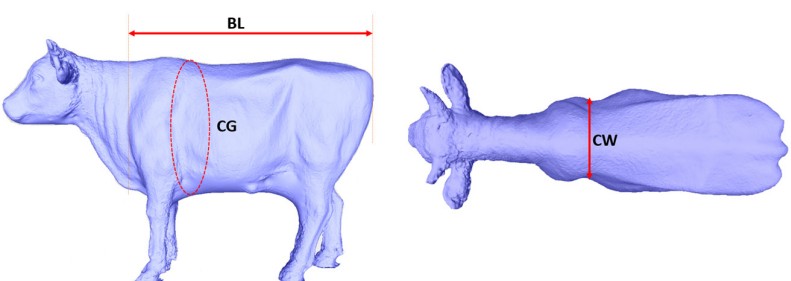

**Figure 6.** Three body dimensions of Korean cattle.

2.3.2. Three-Dimensional Segmentation-Based Feature Extraction

Automatically extracting body dimensions from Korean cattle data acquired in 3D can be a challenging task. To overcome this, we employed a segmentation approach to isolate

the specific parts of the cattle for measurement. We conducted two distinct segmentation processes: one for cattle torso segmentation to measure body length and another for center body segmentation, which allows us to measure chest girth and chest width, as illustrated in Figure 7.

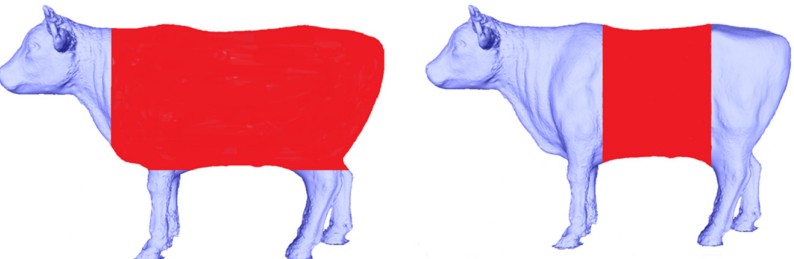

**Figure 7.** Torso segmentation (**left**) and center body segmentation (**right**).

The advancement of artificial intelligence (AI), particularly in deep learning techniques, has introduced powerful tools for 3D data analysis. One such network, PointNet [18], specializes in 3D data analysis and offers the advantage of learning both global and local features. It can be effectively applied to various 3D tasks, including 3D classification, 3D segmentation, and 3D part segmentation. In this project, we adopt the PointNet network for 3D cattle part segmentation. To simplify the data labeling process while maintaining high accuracy, we used binary 3D segmentation, simplifying the model's complexity. Consequently, we implemented two models with identical architecture but distinct labeling data: one for cattle torso segmentation and the other for center body segmentation.

The architectural overview of PointNet, designed for point cloud segmentation tasks, is presented in Figure 8. It incorporates an Input Transform network (T-Net) followed by a series of Multi-Layer Perceptrons (MLPs) for local feature extraction. The Input Transform network captures transformations to ensure the network's robustness to input point permutations, rotations, and translations. Subsequently, a Feature Transform network (T-Net) enhances the network's capacity to handle diverse point orderings. After local feature extraction, a global feature vector is derived through max pooling, enabling the network to aggregate information from the entire point cloud. This global feature vector is further processed by a set of MLPs to produce the final segmentation mask, which assigns class labels to each point. The combination of Input and Feature Transform networks empowers PointNet to effectively segment complex 3D data.

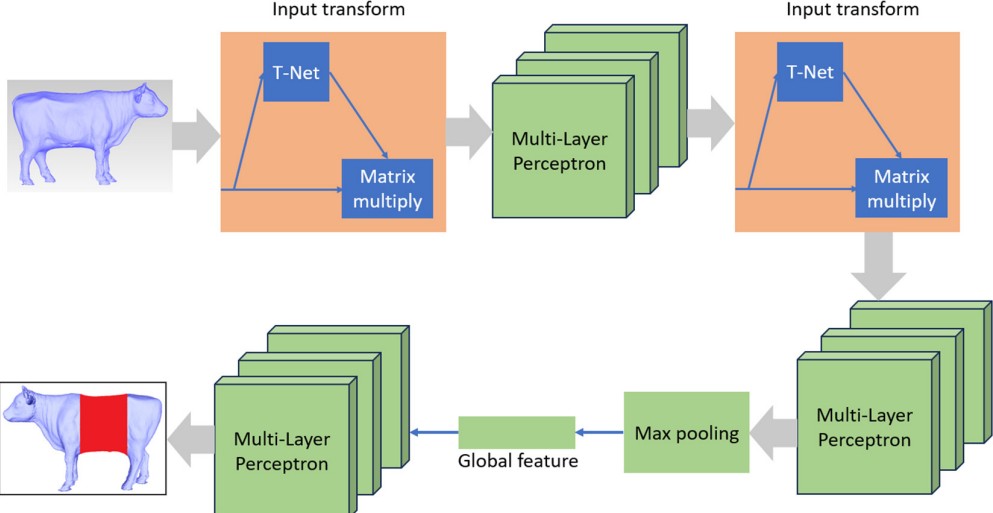

**Figure 8.** PointNet Architecture for 3D segmentation [18].

### 2.4. Regression Machine Learning

With three numeric inputs (body length, chest girth, chest width) and the numeric output of cattle weight, as illustrated in Figure 9, regression models are the most appropriate choice. In this project, we selected five prominent regression machine learning models for Korean cattle weight prediction, CatBoost regression, LightGBM, polynomial regression, random forest regression, XGBoost regression.

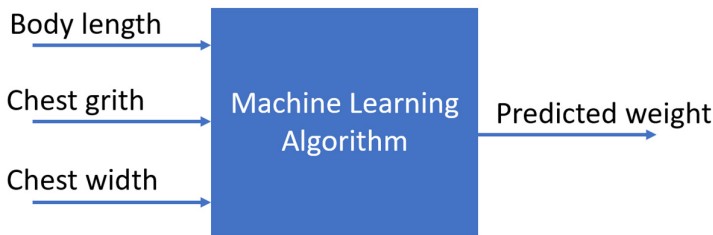

**Figure 9.** Simple schematic of regression machine learning for weight prediction.

### 2.4.1. CatBoost Regression

CatBoost [19] is a renowned ensemble machine learning algorithm, particularly effective in regression tasks. It employs category-based optimization to enhance predictive accuracy and utilizes gradient boosting to iteratively construct decision trees, effectively reducing errors. Notable features of CatBoost include its intrinsic handling of categorical data, adept feature selection, and strategies to prevent overfitting. The algorithm also demonstrates efficiency in real-world applications, offering support for parallel computation and fine-tuned hyper-parameter optimization.

### 2.4.2. Light Gradient Boosting Machine

Grounded in gradient boosting techniques, LightGBM [20] meticulously constructs decision trees to iteratively correct errors. Its innovation lies in histogram-based algorithms and leaf-wise tree growth, ensuring computational efficiency. LightGBM further employs gradient-based one-side sampling and exclusive data filtering to enhance robustness and mitigate overfitting. Its parallel processing capabilities make it an excellent choice for regression tasks.

### 2.4.3. Polynomial Regression

Polynomial regression [21] extends linear regression by incorporating basic mathematical functions. This algorithm is particularly useful for handling nonlinear data by employing linear factors. It demonstrates the capability to work effectively with a wide range of nonlinear data while maintaining efficiency comparable to linear functions.

### 2.4.4. Random Forest Regression

Random forest [22] is an ensemble learning technique based on decision tree models. During training, it builds a collection of decision trees, with each tree constructed independently and accessing a random subset of the training data. The use of random subsets of data and features helps prevent overfitting, contributing to the model's robustness.

### 2.4.5. Extreme Gradient Boost Regression

XGBoost [23] is another ensemble learning technique rooted in the gradient boosting framework, primarily applied to regression tasks. XGBoost iteratively refines predictive models by constructing a series of decision trees, each correcting the errors of the previous iteration. It is distinguished by its incorporation of sophisticated L1 and L2 regularization techniques to mitigate overfitting and maintain model parsimony. XGBoost is also known for its robust handling of missing data.

## 3. Results

### 3.1. Segmentation

3.1.1. Cross-Sampling Augmentation

Building deep learning models always necessitates a substantial number of labeled samples for the training process. In the case of 3D PointNet networks, achieving high model accuracy demands training on thousands of samples. However, the manual labeling of thousands of samples is an exceedingly labor-intensive task. To address this challenge, we introduced an augmentation method named cross-sampling.

The cross-sampling process is illustrated in Figure 10. Starting with each 3D Korean cattle data sample, we conducted down-sampling with a resolution of 0.1 mm. Following down-sampling, each the 3D cattle point cloud typically contains between 11 thousand to 12 thousand points. We partitioned these into N segments (in this study we selected N = 10), each consisting of 1024 points (PointNet with 1024 input was selected for this project). This process yielded N sparse point clouds (depicted in blue in Figure 10) for each original sample. Subsequently, we further divided each sparse point cloud into N segments and recombined them to create additional N samples (depicted in green in Figure 10), distinct from the previous set. Through this approach, with each original 3D mesh data, we generated 2N sparse point cloud samples, each consisting of 1024 points.

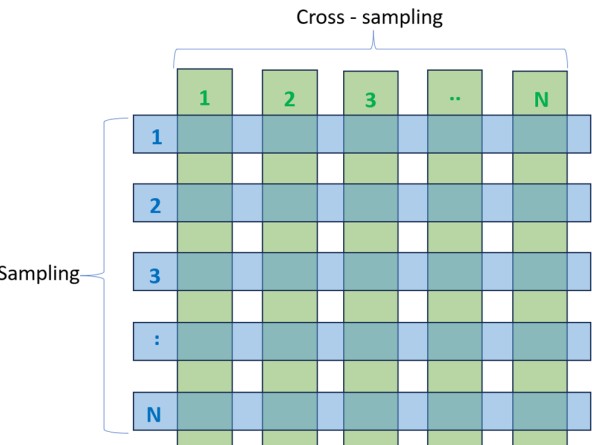

**Figure 10.** Cross-sampling.

The data preparation for the 3D segmentation process was executed as follows: in an effort to reduce the labor-intensive task of manual labeling, we selectively chose 100 files from the total pool of 1190 3D Korean cattle data files to construct 3D segmentation model, with 80 files allocated for the training set and 20 for validation. Subsequently, labeling was performed on these 100 cattle data files, followed by the use of cross-sampling augmentation to expand our dataset by a factor of 20. Consequently, this process yielded 1600 samples for the training set and 400 for the validation set, resulting in a total of 2000 samples.

3.1.2. Feature Extraction

To verify the accuracy of 3D segmentation process, we employed global accuracy metric [24], which is defined as below:

Global accuracy:

$$Global\ Accuracy = \frac{Number\ of\ correct\ prediciton}{Total\ number\ of\ prediction} \tag{1}$$

The experiments were conducted on a computational workstation equipped with a CPU Core-i9 3.5 GHz and an NVIDIA 3060Ti GPU with 8 GB of memory. For deep learning, we chose the TensorFlow 2.1.0 framework [25] and CUDA 11.0. The network parameters included the use of the adaptive moment estimation optimizer (Adam), a batch size of 64,

1000 training epochs, and a learning rate of 0.001. Only the best weights were saved during training. The records of the training history are displayed in Figures 11 and 12, and the accuracy results are summarized in Table 3.

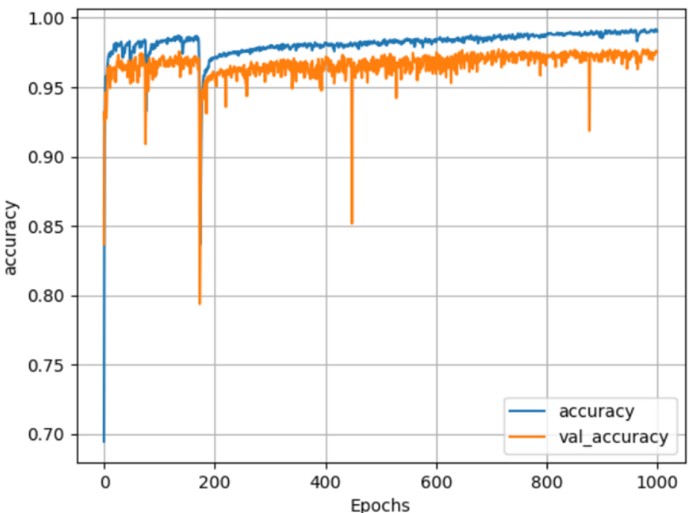

**Figure 11.** Torso segmentation training history plot.

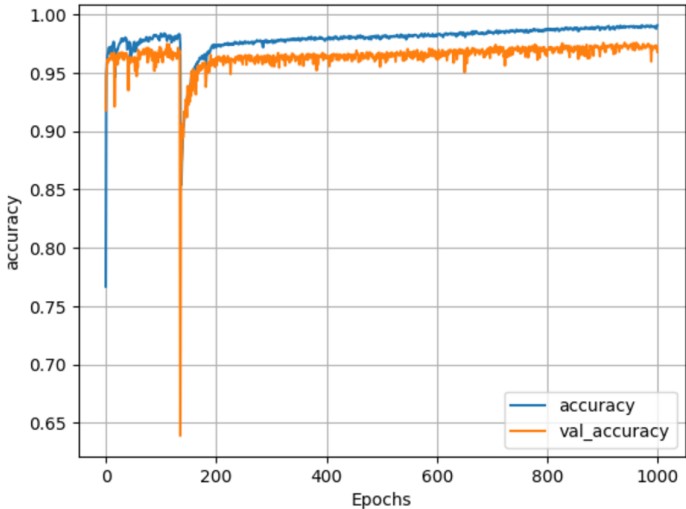

**Figure 12.** Center body segmentation training history plot.

**Table 3.** Three-dimensional segmentation accuracy.

| Case | Training Accuracy | Validation Accuracy |
|---|---|---|
| Torso segmentation | 99.04% | 97.55% |
| Center body segmentation | 99.01% | 97.21% |

In Figures 11 and 12, the blue line represents the training process, while the orange line represents the testing process. The training process was stabilized after approximately 400 epochs, resulting in a training accuracy of 99% and a testing accuracy of 97% for both segmentation cases. We applied the trained segmentation models to perform 3D cattle segmentation, and the results are visualized in Figures 13 and 14.

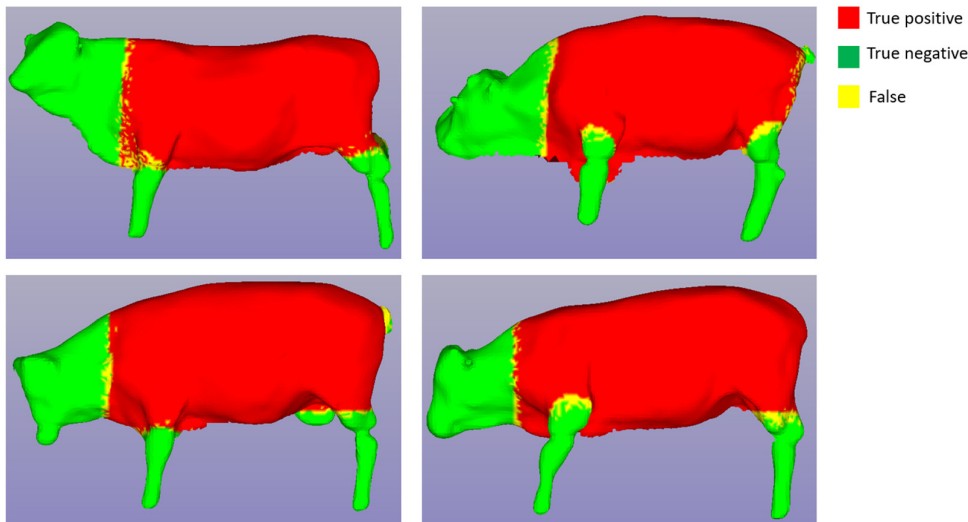

**Figure 13.** Torso segmentation results.

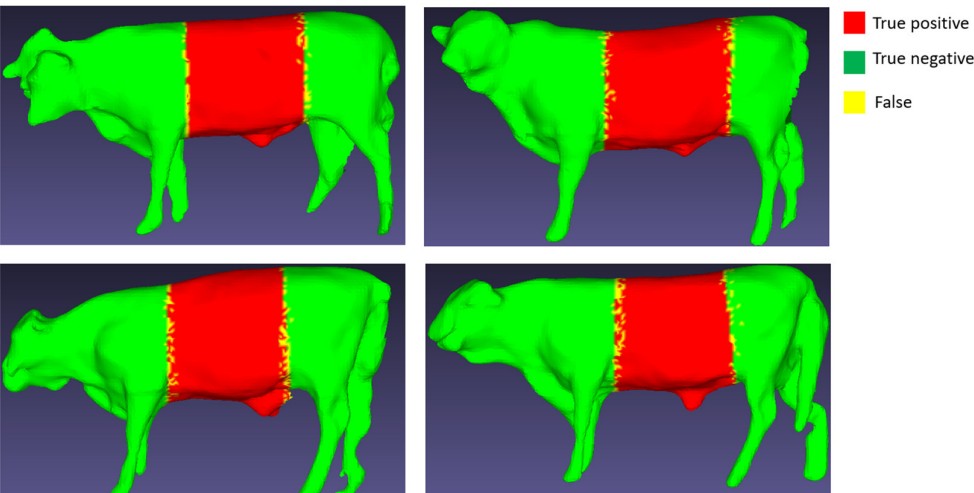

**Figure 14.** Centre body segmentation results.

In Figures 13 and 14, the red area represents "True positive", the green area represents "True negative", and the yellow region indicates False ("False positive" or "False negative"). It is notable that the yellow area occupies very small areas at the border between the red and green areas, which has a negligible impact on the subsequent size measurement.

To achieve accurate measurements of the body dimensions of Korean cattle, it is essential for the cattle to be in an upright position from head to tail. However, in reality, the cattle often stand in a tilted position. To address this, we corrected the animals' posture both horizontally and vertically using rendered silhouettes derived from the 3D-segmented torso.

We employed the principal component analysis (PCA) method [26] for posture correction. The process involved extracting contour points from an image, calculating the centroid of these points to center the data, creating a covariance matrix toestablish the relationship between x and y coordinates, and computing the eigenvalues and eigenvectors of the covariance matrix. The eigenvector with the largest eigenvalue signified the principal axis, aligning with the contour's orientation in both the vertical and horizontal views. The results of orientation correction are illustrated in Figure 15.

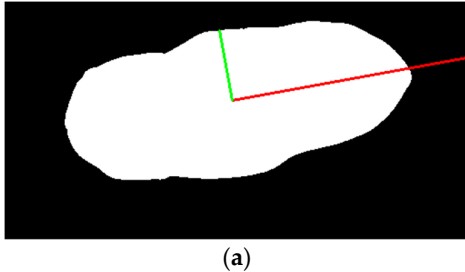 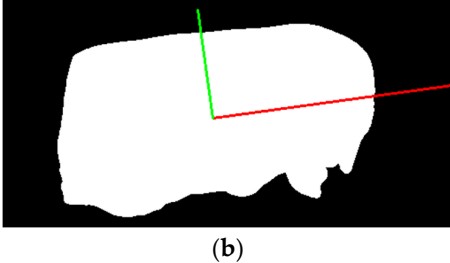

(**a**)  (**b**)

**Figure 15.** Posture correction using PCA: (**a**) top view; (**b**) side view.

Posture correction allowed us to measure body length by capturing the horizontal length of the segmented torso, as shown in Figure 16.

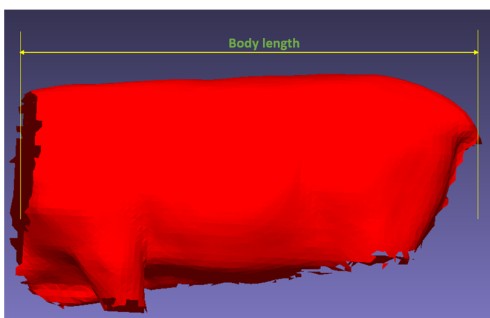

**Figure 16.** Extracting body length from the 3D-segmented torso.

To extract chest girth and chest width, we followed these steps. First, we corrected the animal's posture both horizontally and vertically. Then, we cut planes perpendicular to the animal's body axis to delineate the boundary surrounding its chest. Despite not obtaining a closed contour due to the limitations of the 3D data collection system, the achieved contour encompassed over 60% of the cattle's chest, facilitating the interpolation of a ellipse. We fitted an ellipse to the achieved contour, with the perimeter of the fitted ellipse measuring chest girth and the minor axis of the ellipse measuring chest width. Figure 17 on the left displays a 3D image of cattle after center body segmentation, and Figure 17 on the right depicts the extraction process.

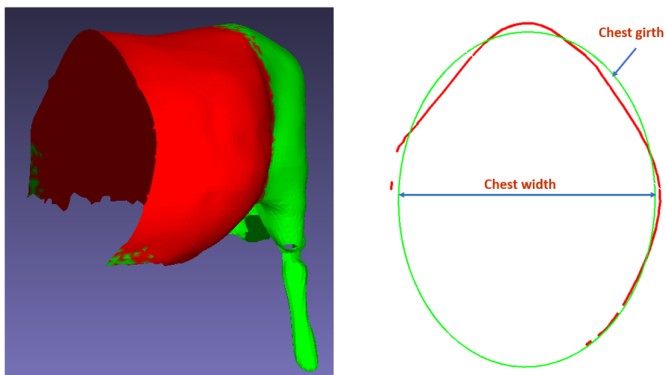

**Figure 17.** Extracting chest girth and chest width from segmented center body.

*3.2. Weight Prediction*

Having extracted the three dimensions from 1190 3D Korean cattle samples in the previous step, we investigated the relationship between body sizes and cattle weight within the dataset. We utilized Pearson's correlation [27] coefficient to calculate linear correlation between features and weight. Pearson's correlation *R* can be observed in Equation (2).

Pearson's correlation coefficient:

$$R = \frac{\sum_{i=1}^{n}(x_i - \overline{x})(y_i - \overline{y})}{\sqrt{\sum_{i=1}^{n}(x_i - \overline{x})^2}\sqrt{\sum_{i=1}^{n}(y_i - \overline{y})^2}} \qquad (2)$$

where:

$R$ is the correlation coefficient.

$x_i$ and $y_i$ are the values of the x-variable and y-variable.

$\overline{x}$ and $\overline{y}$ are the means of the values of the *x*-variable and *y*-variable, respectively.

Figure 18 presents a scatter plot showcasing the relationship between the cattle weight and each dimension, along with correlation coefficients calculated between weight and body length, chest girth, and chest width, respectively: 0.632, 0.524, and 0.428. These coefficients represent varying degrees of the positive linear relationship between the animal's weight and each dimension. Specifically, there appears to be high correlations with body length and chest girth, while the correlation with chest width is weaker.

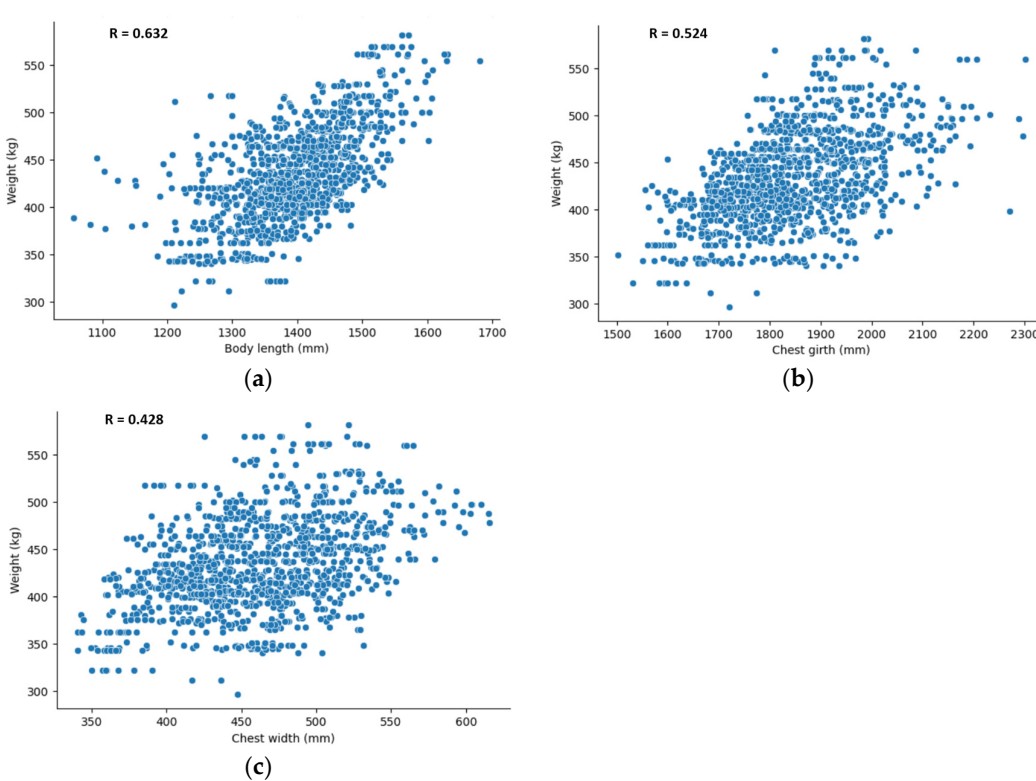

**Figure 18.** Scatter plot of the relationship between three body dimensions and weight of Korean cattle: (**a**) body length and weight; (**b**) chest girth and weight; (**c**) chest width and weight.

### 3.2.1. K-Fold Cross-Validation

To assess the performance of the chosen machine learning models, we employed K-fold cross-validation. The data were randomly divided into ten partitions of equal size (k = 10). For each partition (p), we trained the selected machine learning models on the remaining nine partitions and subsequently tested the models on partition (p). The final score was computed as the average of all ten scores obtained. The schematic of K-fold validation with k = 10 is depicted in Figure 19.

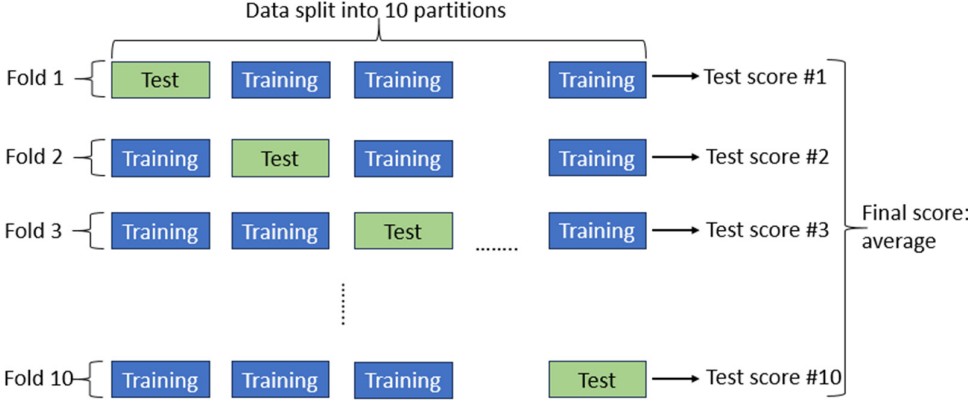

**Figure 19.** Schematic of K-fold cross-validation with k = 10.

3.2.2. Evaluation Metrics

To evaluate weight prediction performance of the proposed approach, two standard evaluation metrics are used. We used the mean absolute error (MAE) and mean absolute percentage error (MAPE).

MAE:

$$MAE = \frac{1}{n}\sum_{i=1}^{n}|yi - pi| \tag{3}$$

where:

$n$ is a number of tested samples.

$yi, i = 1, \ldots, n$ is a known value of cattle weight.

$pi, i = 1, \ldots, n$ is a predicted value of the cattle weight.

MAPE:

$$MAPE = \frac{1}{n}\sum_{i=1}^{n}\left|\frac{yi - pi}{yi}\right| \tag{4}$$

where:

$n$ is a number of tested samples.

$yi, i = 1, \ldots, n$ is a known value of cattle weight.

$pi, i = 1, \ldots, n$ is a predicted value of the cattle weight.

3.2.3. Results and Discussion

The experiments aimed to estimate Korean cattle weight using the five proposed machine learning models: CatBoost regression, LightGBM, polynomial regression, random forest regression, and XGBoost regression. These experiments were conducted ten times. The results are displayed in Tables 4–8, and the average performance across the ten experiments is summarized in Table 9.

**Table 4.** CatBoot regression result.

| Fold Number | Evaluation Metrics | |
|---|---|---|
| | MAE (kg) | MAPE (%) |
| Fold 1 | 27.800 | 6.529 |
| Fold 2 | 27.116 | 6.320 |
| Fold 3 | 27.767 | 6.380 |
| Fold 4 | 25.776 | 6.014 |
| Fold 5 | 26.432 | 6.371 |
| Fold 6 | 26.164 | 6.031 |
| Fold 7 | 25.775 | 5.980 |

**Table 4.** *Cont.*

| Fold Number | Evaluation Metrics | |
| :---: | :---: | :---: |
| | **MAE (kg)** | **MAPE (%)** |
| Fold 8 | 26.572 | 6.175 |
| Fold 9 | 29.491 | 6.924 |
| Fold 10 | 25.296 | 5.880 |
| **Average** | **26.819** | **6.260** |

**Table 5.** LightGBM regression result.

| Fold Number | Evaluation Metrics | |
| :---: | :---: | :---: |
| | **MAE (kg)** | **MAPE (%)** |
| Fold 1 | 26.268 | 6.124 |
| Fold 2 | 24.656 | 5.712 |
| Fold 3 | 26.193 | 6.094 |
| Fold 4 | 24.284 | 5.731 |
| Fold 5 | 25.383 | 6.033 |
| Fold 6 | 26.272 | 6.045 |
| Fold 7 | 24.096 | 5.537 |
| Fold 8 | 25.042 | 5.844 |
| Fold 9 | 26.560 | 6.191 |
| Fold 10 | 26.760 | 6.173 |
| **Average** | **25.551** | **5.948** |

**Table 6.** Polynomial regression result.

| Fold Number | Evaluation Metrics | |
| :---: | :---: | :---: |
| | **MAE (kg)** | **MAPE (%)** |
| Fold 1 | 25.302 | 5.903 |
| Fold 2 | 26.085 | 6.078 |
| Fold 3 | 26.187 | 6.066 |
| Fold 4 | 24.871 | 5.805 |
| Fold 5 | 25.594 | 6.116 |
| Fold 6 | 23.714 | 5.433 |
| Fold 7 | 25.017 | 5.790 |
| Fold 8 | 25.301 | 5.858 |
| Fold 9 | 27.933 | 6.527 |
| Fold 10 | 26.233 | 6.080 |
| **Average** | **25.624** | **5.966** |

**Table 7.** Random forest regression result.

| Fold Number | Evaluation Metrics | |
| :---: | :---: | :---: |
| | **MAE (kg)** | **MAPE (%)** |
| Fold 1 | 25.256 | 5.903 |
| Fold 2 | 24.293 | 5.649 |
| Fold 3 | 26.749 | 6.181 |
| Fold 4 | 25.786 | 5.994 |
| Fold 5 | 24.264 | 5.755 |
| Fold 6 | 24.318 | 5.559 |
| Fold 7 | 24.955 | 5.682 |
| Fold 8 | 25.294 | 5.890 |
| Fold 9 | 26.856 | 6.282 |
| Fold 10 | 24.269 | 5.622 |
| **Average** | **25.204** | **5.852** |

**Table 8.** XGBoost regression result.

| Fold Number | Evaluation Metrics | |
|---|---|---|
| | **MAE (kg)** | **MAPE (%)** |
| Fold 1 | 27.257 | 6.393 |
| Fold 2 | 27.150 | 6.285 |
| Fold 3 | 27.359 | 6.277 |
| Fold 4 | 27.252 | 6.370 |
| Fold 5 | 26.066 | 6.188 |
| Fold 6 | 27.299 | 6.296 |
| Fold 7 | 25.569 | 5.830 |
| Fold 8 | 27.052 | 6.274 |
| Fold 9 | 28.070 | 6.538 |
| Fold 10 | 26.401 | 6.108 |
| **Average** | **26.948** | **6.256** |

**Table 9.** Average results.

| Model | Evaluation Metrics | |
|---|---|---|
| | **Average of MAE (kg)** | **Average of MAPE (%)** |
| CatBoost regression | 26.819 | 6.260 |
| LightGBM regression | 25.551 | 5.948 |
| Polynomial regression | 25.624 | 5.966 |
| Random forest regression | 25.204 | 5.852 |
| XGBoost regression | 26.948 | 6.256 |

The average MAE and MAPE for these algorithms are visualized in Figures 20 and 21. As depicted in the bar graphs, the random forest model exhibited the highest performance, achieving an MAE error of 25.2 kg and a MAPE of 5.852%. It was followed by LightGBM and polynomial regression. XGBoost and CatBoost also performed well among the models.

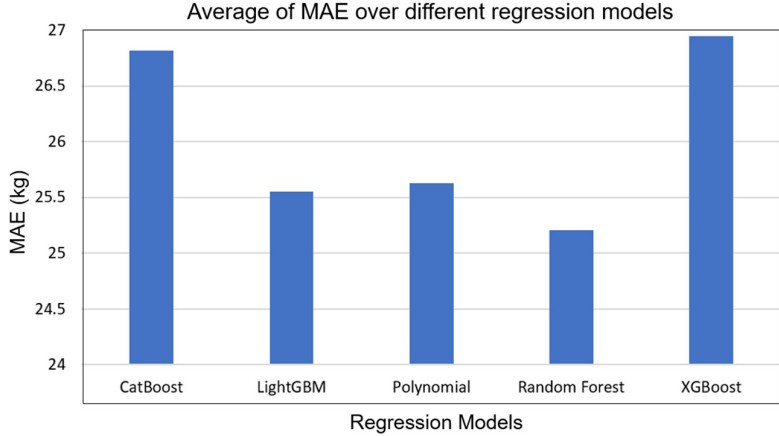

**Figure 20.** Average MAE results of 10 fold experiments.

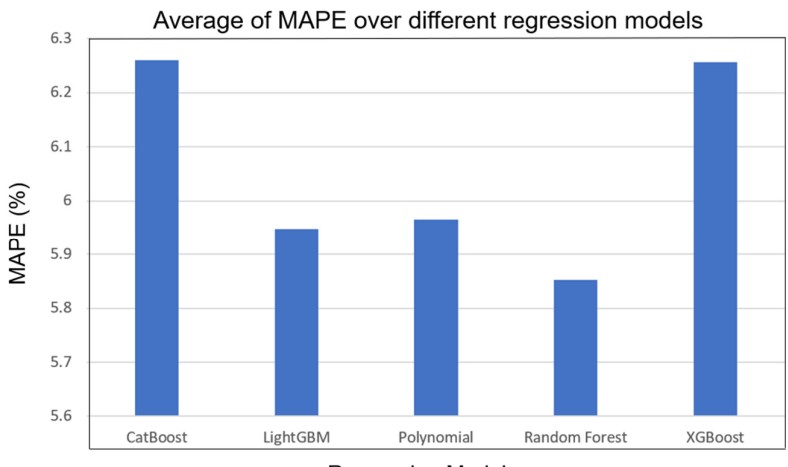

**Figure 21.** Average MAPE results of 10 fold experiments.

To assess the effectiveness of estimating Korean cattle weight using the proposed dimensions, we analyzed the estimation results in comparison to the actual cattle weight for each machine learning model. We calculated the correlation coefficients between the predicted weight and the actual weight for each model by employing Equation (2). Figure 22 presents these results, where the correction coefficients show the strengths of the linear relationship between the predicted and actual weights across all models. Notably, random forest and LightGBM demonstrate marginally stronger correlations in comparison to the remaining models. Conversely, XGBoost displays a slightly lower correlation but is still significant in magnitude.

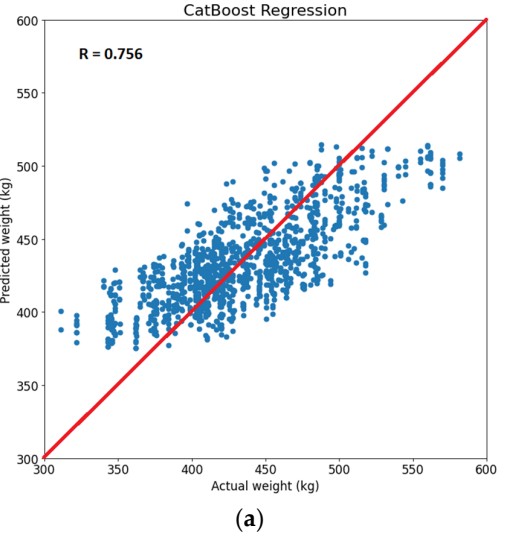

(**a**)

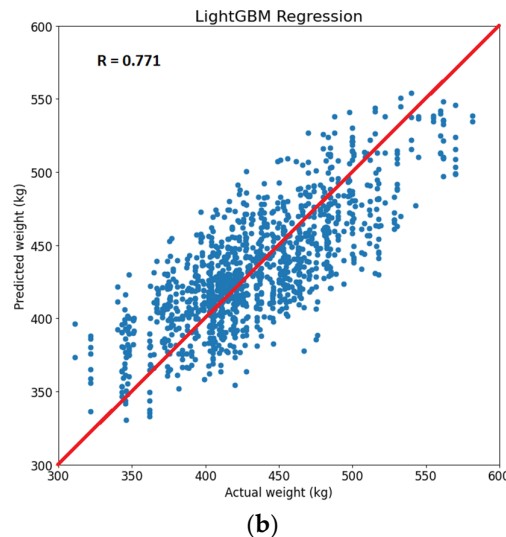

(**b**)

**Figure 22.** *Cont.*

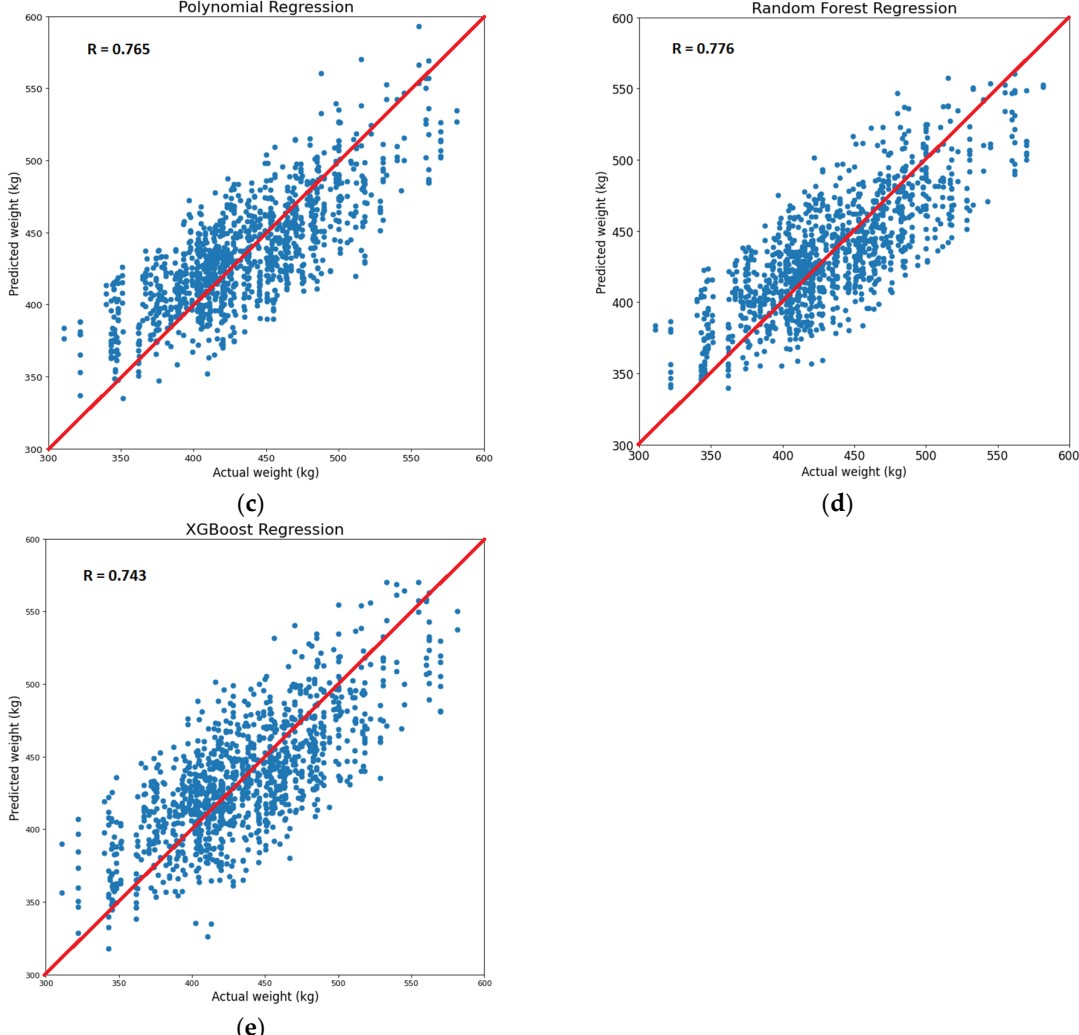

**Figure 22.** Scatter plot of predicted weight values and measured weight values and the correlation coefficient (R) on different regression machine learning models: (**a**) CatBoost regression; (**b**) LightGBM regression; (**c**) polynomial regression; (**d**) random forest regression; (**e**) XGBoost regression.

Table 10 presents a performance comparison between our approach and previous methodologies. We compared our results with the studies that also experimented on cattle and used MAPE or MAE as performance metrics. In terms of MAPE, our approach outperforms all other methods. Regarding MAE, this metric is used only in the studies by Ruchay et al. [10] and Weber et al. [4]. Our MAE demonstrates superiority over the findings in [10]. Although [4] shows a better MAE compared to ours, it is important to note that their experiments were conducted in a feeding fence system, whereas our experiments operated in real farm environments.

In this study, although the collected data exhibited limitations, the missing information from regions on the right side and the under area of the cattle can be attributed to the adaptation of our design to real farm environmental conditions. Additionally, our model was constructed using only three body dimensions, yet it demonstrated significant efficacy in predicting the weight of Korean cattle. In future iterations, we aim to enhance our predictive results by incorporating additional dimensions such as rump width and hip height. This expansion is anticipated to bolster the accuracy of our weight prediction solution.

**Table 10.** Comparison with previous publications.

| No. | Work | Cattle Number | Environment | MAPE (%) | MAE (kg) |
|---|---|---|---|---|---|
| 1 | Jang et al. [5] | 209 | Real farm | 19.10 | - |
| 2 | Anifah and Haryanto [2] | 13 | Fence system | 18.76 | - |
| 3 | Yamamoto et al. [28] | 105 | Real farm | 12.45 | - |
| 4 | Ruchay et al. [10] | 275 | In door | 9.60 | 40.10 |
| 5 | Nishide et al. [29] | 184 | Cattle barn | 6.39 | - |
| 6 | Weber et al. [4] | 110 | Feeding fence | - | **13.44** |
| 7 | **Proposed approach** | **270** | **Real farm** | **5.85** | **25.20** |

The data utilized in this study is entirely collected from real farm environments, ensuring the study's outcomes closely mirror practical scenarios, allowing for potential real-world deployment with minimal differences. There are two differences when deploying our algorithm in real world in comparison to our experiments. The first difference is cattle movement, and the second one is that the cattle may stand in unexpected poses involving tilting, turning, or bowing their heads, which may affect precise body dimension extraction. To address the cattle movement problem, a triggering mechanism synchronizing 10 cameras was employed, ensuring the simultaneous capture to create stationary cattle images. To address the unwanted posture problem, we developed a preprocessing algorithm capable of discerning favorable cattle positions, enabling the exclusion of frames with undesirable poses. We only took images when good cattle positions were confirmed. The remaining processes are identical to the experiments described in this work.

Our approach not only demonstrates the efficacy of weight prediction for Korean cattle but also presents potential applicability to other species of similar size, including dairy cows and various type of cattle. From a solution-oriented perspective, this methodology can be adapted to automatically measure the weight of other animals, such as sheep and pigs, by making slight adjustments to the mechanical size of the multiple-camera system. In these cases, data collection and model construction would need to follow a process similar to ours for effective implementation.

## 4. Conclusions

In this paper, we presented a vision-based solution for predicting the weight of Korean cattle using 3D segmentation and regression machine learning. After acquiring data using multi-camera system, we employed PointNet for 3D segmentation, conducting two distinct experiments: one to segment the torso for extracting body length and the other to segment the center body for extracting chest girth and chest width. Finally, we applied five machine learning algorithms to estimate cattle weight based on the three extracted body dimensions. We conducted experiments on 1190 3D Korean cattle samples, captured from various poses of 270 Korean cattle. The results of these experiments demonstrated an accuracy of 25.2 kg in terms of MAE and 5.85% in terms of MAPE. Our approach not only showcases the effectiveness of weight prediction for Korean cattle but also holds the potential for broader applicability to other species.

**Author Contributions:** Conceptualization, C.G.D. and S.H.; methodology, S.H.; software, S.H., M.K.B. and V.T.P.; validation, S.S.L., M.A. and S.M.L.; formal analysis, M.A.; investigation, C.G.D.; resources, M.N.P.; data curation, H.-S.S.; writing—original draft preparation, S.H.; writing—review and editing, S.H., C.G.D. and J.G.L.; visualization, M.K.B. and V.T.P.; supervision, C.G.D.; project administration, C.G.D.; funding acquisition, C.G.D. All authors have read and agreed to the published version of the manuscript.

**Funding:** This research was funded by the Korea Institute of Planning and Evaluation for Technology in Food, Agriculture and Forestry: 421050-03 and the Korea Smart Farm RD Foundation (KosFarm) through Smart Farm Innovation Technology Development Program, funded by the Ministry of

Agriculture, Food and Rural Affairs (MAFRA) and the Ministry of Science and ICT (MSIT), Rural Development Administration (421050-03).

**Institutional Review Board Statement:** The animal study protocol was approved by the IACUC at National Institute of Animal Science (approval number: NIAS 2022-0545/approval date: 12 May 2022).

**Data Availability Statement:** The datasets generated during this study are available from the corresponding author upon request.

**Conflicts of Interest:** The authors, Min Ki Baek, Van Thuan Pham, Seungkyu Han were employed by the company ZOOTOS Co., Ltd. The remaining authors declare that the research was conducted in the absence of any commercial or financial relationships that could be construed as a potential conflict of interest.

## Abbreviations

The following abbreviations are used in this paper.

| | |
|---|---|
| 2D | Two Dimensional |
| 3D | Three Dimensional |
| AI | Artificial Intelligence |
| ANN | Artificial Neural Network |
| MAE | Mean Absolute Error |
| MAPE | Mean Absolute Percentage Error |
| MLP | Multiple Layer Perceptron |
| LIDAR | Light Detection and Ranging |
| PCA | Principal Component Analysis |
| RGB-D | Red Green Blue Depth |
| LightGBM | Light Gradient Boosting Machine |
| XGBoost | Extreme Gradient Boost |

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
