# Peer review of "A Korean Cattle Weight Prediction Approach Using 3D Segmentation-Based Feature Extraction and Regression Machine Learning from Incomplete 3D Shapes Acquired from Real Farm Environments"

_agriculture, doi:10.3390/agriculture13122266_

Round 1
Reviewer 1 Report
Comments and Suggestions for Authors
The paper addresses the interesting problem of the weight estimation of the cattle using the reconstructed stereo images. The paper is well written, and the details of the procedure for creating the dataset are well explained. Then, using the measurements of the torso and chest, the work emphasizes the use of regression models to predict the weight. The following are the points that need to be addressed:
- The MAE of 25.2 kg is higher; how does it impact the overall efficacy of the proposed weight prediction method?
- In the reconstructed mesh, there are some of the missing regions. How does it impact the overall accuracy of the model? What is the margin of error considered in the proposed reconstruction?
- Can any other models be used instead of the regression ones for weight estimation, and how effective are they in predicting the weight?
- The camera set being used; how effective is it in dealing with variation in the height of the cattle? Also, the paper misses the specs of the cameras used.
- How can the proposed system be further expanded for some other animal species?
- Show how different factors, such as data quality, feature selection, and model complexity, can affect the accuracy of weight estimation models based on regression techniques.
- Also, try to evaluate the performance of the proposed work with other weight estimation methods only then the efficacy of the proposed method can be validated.
- Details are also requested on how the results are verified in the real world.
Reviewer 2 Report
Comments and Suggestions for Authors
In this study, the authors have presented a vision-based solution for predicting the weight of Korean cattle using 3D segmentation and regression machine learning. After acquiring data from the multi-camera system, PointNet was used for 3D segmentation, conducting two distinct cases: one to segment the torso for extracting body length and another to segment the center body for extracting chest girth and chest width. Finally, five machine learning algorithms were applied to estimate cattle weight based on the three extracted dimensions. Overall, this research has good innovation and practicability, and can improve the effectiveness of weight prediction for Korean cattle. Moreover, the following issues need to be addressed.
(1) Line 162: The expression of this sentence is not appropriate. The five models used in this paper are traditional machine learning methods that are not the most advanced.
(2) Line 167-172: Why choose the three parameters (body length, chest girth, chest width) out of ten parameters, and can the system measure more parameters to improve prediction accuracy?
(3) Line 300: In the description of 3D segmentation results, there is a lack of comparative analysis with the results of other existing literatures.
(4) Line 301-302: The division of the training set and the test (validation) set is not described in the segmentation experiment.
(5) Line 347-348: The correlation between weight and the three features (body length, chest girth, chest width) can be analyzed respectively, and the correlation coefficient R can be calculated.
(6) Line 376: In the description of weight prediction results, there is a lack of comparative analysis with the results of other existing literatures.
(7) Line 383-386: The styles in Figure 20 and 21 need to be adjusted, such as adding axes.
(8) Line 393-395: In each model, the correlation between the predicted weight and the actual weight can be analyzed, and the correlation coefficient R can be calculated.
(9) Line 405: The data is inconsistent with the data in Table 8.
Reviewer 3 Report
Comments and Suggestions for Authors
A Korean Cattle Weight Prediction Approach Using 3D Segmentation-Based Feature Extraction and Regression Machine Learning from Incomplete 3D Shapes Acquired from Real Farm Environments
The paper proposes a method based on 3D segmentation feature extraction and regression machine learning to predict the weight of Korean Cattle. The method utilizes 3D data of Korean Cattle obtained from real farm environments and employs the Point Net network model, based on deep learning, for 3D segmentation. Five regression machine learning models are used for weight prediction. The paper analyzes multiple metrics, conducts various model experiments and provides visualizations to demonstrate the advantages and feasibility of the proposed method for weight prediction. However, the paper does not improve the main network module; it solely focuses on model comparisons. Moreover, the Point Net network model, which was introduced several years ago, could be replaced with newer backbone networks and models such as Point Net++ or KPConv, which might offer better detection accuracy and speed. Additionally, the paper contains several errors in detail expression, including vague language, inconsistent text formatting, and layout mistakes.
Shortcomings and suggestions:
1. The first occurrence of English abbreviations requires a full English name.
2. The article cites relevant literature on "2D image analysis and 3D image analysis", but lacks a summary analysis of the literature.
3. Figure 4. In the weight distribution of Korean cattle used in this study, there is an imbalance in the weight distribution data provided.
4. Lines 279-281 indicate that the dataset has been expanded through data augmentation. It is recommended to display the expanded dataset through charts. Has the data augmentation solved the problem of data imbalance in the previous text.
5. The training accuracy and validation accuracy shown in Table 2 do not provide the proportion of dataset partitioning.
6. The manuscript did not make any improvements to the backbone network module, but only compared the models. Moreover, the PointNet network model has been proposed for some years, and it can be attempted to replace with newer backbone networks and models, such as PointNet++and KPConv models, which may achieve better detection accuracy and speed.
7. Please provide a detailed analysis of where the selected variables of chest circumference, chest width, body length, and weight are reflected in the scatter plot explanation in the "weight prediction" section.
Comments on the Quality of English Language
Disadvantages and Suggestions:
1. Pay attention to Spaces/punctuation between words.
2. The first occurrence of abbreviations needs full English names.
3. Omission or incorrect use of items
Round 2
Reviewer 1 Report
Comments and Suggestions for Authors
The authors have addressed all the comments.
Comments on the Quality of English Language
It requires minor editing.
Reviewer 2 Report
Comments and Suggestions for Authors
In this study, the authors have presented a vision-based solution for predicting the weight of Korean cattle using 3D segmentation and regression machine learning. After acquiring data from the multi-camera system, PointNet was used for 3D segmentation, conducting two distinct cases: one to segment the torso for extracting body length and another to segment the center body for extracting chest girth and chest width. Finally, five machine learning algorithms were applied to estimate cattle weight based on the three extracted dimensions. Overall, this research has good innovation and practicability, and can improve the effectiveness of weight prediction for Korean cattle. In the revised manuscript, all problems in my comments have been addressed. The manuscript can be accepted in present form.
Reviewer 3 Report
Comments and Suggestions for Authors
The authors have addressed and explained all previous comments. I
suggest accepting their advice.